# Autonomous self-healing organic crystals for nonlinear optics

**Saikat Mondal** [1], **Pratap Tanari** [1,4], **Samrat Roy** [2,4], **Surojit Bhunia** [1], **Rituparno Chowdhury** [1], **Arun K. Pal** [3], **Ayan Datta** [3], **Bipul Pal** [2] ✉ & **C. Malla Reddy** [1] ✉

Non-centrosymmetric molecular crystals have a plethora of applications, such as piezoelectric transducers, energy storage and nonlinear optical materials owing to their unique structural order which is absent in other synthetic materials. As most crystals are brittle, their efficiency declines upon prolonged usage due to fatigue or catastrophic failure, limiting their utilities. Some natural substances, like bone, enamel, leaf and skin, function efficiently, last a lifetime, thanks to their inherent self-healing nature. Therefore, incorporating self-healing ability in crystalline materials will greatly broaden their scope. Here, we report single crystals of a dibenzoate derivative, capable of self-healing within milliseconds via autonomous actuation. Systematic quantitative experiments reveal the limit of mechanical forces that the self-healing crystals can withstand. As a proof-of-concept, we also demonstrate that our self-healed crystals can retain their second harmonic generation (SHG) with high efficiency. Kinematic analysis of the actuation in our system also revealed its impressive performance parameters, and shows actuation response times in the millisecond range.

Multifunctional adaptive dynamic crystalline materials with ability to self-repair[1–3], self-actuate[1], or with mechanical adaptability[4], hold great potential for uses in smart technologies such as in flexible devices[5], soft-robotics[6], energy[7–9], optical[5,10–12], electrical[13], bio-medical[14], actuators[15–19], and high precision sensors[20,21]. Despite this, autonomous self-healing – the ability of a material to recover and restore its pristine properties without the need of any external intervention[22] – is poorly understood in synthetic materials[23], and even more so in crystalline[1–3] and hard materials[1], which is highly desirable to meet future needs of high-end technologies[24–26]. It is not always viable to repair components, such as in a space equipment or micro-devices deployed in remote operations. Hence discovery of new ways of incorporating self-healing property in synthetic materials has become critical for overcoming current challenges. On the other hand, nature found many ways to achieve self-healing[27,28]. For instance, human skin self-heals via

an inflammatory response of cells below the dermis by increasing collagen production, while bone uses electrical potentials generated at the crack junction in the healing process[29].

Over the last two decades, a handful of excellent physical and chemical methodologies have been developed, mainly exploiting diffusion in materials[23,24], shape-memory effect[23], reshuffling ability of covalent bonds[3,25], supramolecular dynamics[24,30,31], or a combination of these[23]. Self-healing in materials almost always requires long contact periods and external stimuli like heat[24], mechanical pressure[3,30], light[31], or a chemical agent[32], which makes the real-life usability of self-healing materials challenging.

Studies on self-healing materials in the literature mainly focus on restoring mechanical properties (toughness, stiffness, etc.), such as in gels[24–26], polymer films[30,31], and composites[32] in which the mechanical strength is a crucial aspect. On the other hand, most known examples

[1]Department of Chemical Sciences, Indian Institute of Science Education and Research Kolkata, Nadia 741246 West Bengal, India. [2]Department of Physical Sciences, Indian Institute of Science Education and Research Kolkata, Nadia 741246 West Bengal, India. [3]School of Chemical Sciences, Indian Association for the Cultivation of Science, Kolkata 700032 West Bengal, India. [4]These authors contributed equally: Pratap Tanari, Samrat Roy. ✉e-mail: bipul@iiserkol.ac.in; cmallareddy@gmail.com

also belong to amorphous or semi-crystalline soft materials[23]. In case of single crystalline materials, which have numerous aforementioned applications[5–21], strategies that aid in material healing with regard to their internal order will be crucial for maintaining their performances[1]. For instance, utility of the piezoelectric[7,8,33,34], ferroelectric[8,35,36], or second harmonic generator materials[37–42] vastly depends on preservation of their non-centrosymmetric, structural order. SHG active materials have enormous applications in laser technologies[43], biomedical[44], and telecommunications[45]. The realization of self-healing in molecular crystals would also address their typical fragile nature, as recently reported by Naumov and coworkers[3,46]. However, self-healing in organic materials with crystallographic precision, which is highly desirable to expand their scope for many applications, remains a great challenge[2,3,47] and the examples in this direction are surprisingly minuscule in number.

Here, we report single crystals of an easily accessible organic compound, dimethyl-4,4'-(methylenebis(azanediyl))dibenzoate (hereafter, **1**), which shows ultrafast mechanical actuation and exceptional autonomous self-healing to recover from mechanically induced cracks of as large as tens of micrometers, without any need for external stimuli. The crystals of **1** remain stable for more than a year at ambient conditions and show high thermal stability upto ~ 200 °C (Supplementary Fig. 1). Systematic mechanical fracture experiments performed on the crystals allowed us to quantify the threshold of load within which the crystals autonomously self-heal. According to statistical data, there is a clear limit to the amount of load that crystals of **1** may sustain before failing to heal themselves completely. Remarkably, the healing events occur in a millisecond timescale, as opposed to hours to days needed in case of most other self-healing materials. Further the fractured crystals recombine with an ultrafast mechanical motion. Such automated dynamics have never been observed in other types of crystalline actuator materials. Hence, we also explored the autonomous actuation motion in this work by studying performance parameters and actuation time. Actuators, which undergo different levels of mechanical agitation, inevitably experience mechanical damages. Hence, the fabricated parts/pieces loosen or fall apart. Therefore, bringing them together will call for outside assistance. Here we show that, the fragmented shards of crystal **1** attract one another without any external aid and recombine autonomously. We also numerically demonstrate that the actuation performances of these crystals are comparable to some real-world actuator classes (MEMS & electroactive polymers). Additionally, exploiting the non-centrosymmetric nature of these crystals, we investigated the nonlinear optical response, i.e., second harmonic generation (SHG) activity. Our SHG studies show an impressive self-healing efficacy in the healed regions of the crystal, which remain comparable to the unaffected regions. Hence, we believe, this material, we report here with several notable findings provides an advantage over conventional SHG active materials as it is not only SHG active but also capable of self-healing instantly via ultrafast actuation.

## Results

### Demonstration of self-healing
Needle-shaped single crystals of **1** were obtained by slow evaporation method from methanol solution in three to four days under ambient conditions (see Fig. 1a, b, c; Methods).

Clean, defect-free single crystals with dimensions of about 1 mm × 0.07 mm × 0.05 mm were picked and subjected to three-point bending tests on a glass slide support under a stereo-microscope equipped with a high-speed camera (Fig. 1f, g). Crystals of **1**, when excessive mechanical stress is applied, fracture in a brittle manner without any sign of noticeable plastic deformation at the macroscopic scale (Fig. 1d, f). If the force is sufficiently high and applied abruptly, the fractured shards snap away or may leave an uneven fracture surface with debris, making the self-healing inefficient. On application of a

gentle force, the crystals generally broke with linear cracks (Fig. 1e, g). Upon withdrawal of the force, the broken fragments recombine due to a significant attractive force that helps in the rejoining process. We observed a perfect healing with no sign of cracks in cases when the two broken ends self-align and close the gap with precision (Fig. 1g, Supplementary Fig. 2 and Supplementary Movies 1–5). Such perfectly-healed crystals appear remarkably similar to the pristine ones under optical microscope. It is to be noted that in hard and crystalline materials reported in literature, the self-healing is mostly observed at the nanoscale[48] or subtly at the micrometer scale and examples of perfect healing at macroscopic scale hardly exist. We observed that the landing of pieces in a perfect manner was challenging due to the fast attractive motion of the broken pieces or uneven generation of fracture surfaces that often lead to misalignment. Even in such cases, where a visible crack-line remain at the joint (Supplementary Fig. 3) (hereafter, imperfectly healed crystals), the crystals were still capable of holding together and behaved like a monolithic sample.

The mechanical fracture tests on the crystals of **1** also show that they develop long-lasting opposite charges on broken surfaces, which help in the recombination of fractured shards upon withdrawal of the external mechanical stress similar to what we observed earlier in another study[1]. The electrical charges that develop on fractured surfaces[1,49] (Supplementary Fig. 4) form the basis for self-actuation and healing in crystals of **1**.

### Quantification of mechanical force for perfect self-healing
To demonstrate the healing potential of our crystals, we further repeated the experiments by applying uniaxial stress on crystals fixed on a flat support (see Methods, Fig. 2a, b and Supplementary Movies 6, 7). We used a force sensor to quantitatively distinguish the loads required for perfect and imperfect self-healing events (often, the latter is associated with complete fracture of the crystals due to excessive damage). The force sensor with a spherical tip having a diameter of ~190 micrometers was used for compressing the crystals with thicknesses ranging between ~70 and 100 μm. As the crystals were fixed on the substrate in this method, we achieved a better control in the experiments and could repeatedly apply load in cycles on each crystal to observe multiple healing cycles (see the Supplementary Movies 6, 7).

Our exhaustive statistical analysis of the fracture tests using this method show that the load required to fracture crystals and achieve perfect self-healing is within milli Newton (mN) range (see Fig. 2c, Supplementary Table 1) for crystals of thickness ranging between ~70 to 100 μm. The loads about ~ 70 mN and beyond mostly resulted in imperfect healing or permanent deformation as this was sufficient to crush the crystals, leaving the material with severe fracture. Nevertheless, using the gentle uniaxial compression stress (typically the loads being <70 mN), we could show excellent repeatable healing cycles, as high as up to 10 cycles (see Supplementary Movie 8). The controlled uniaxial compression experiments demonstrate that the crystals have an ability to withstand external mechanical stresses repeatedly, as long as the sustainable limit is not crossed.

### Healing timescale
Short healing timescales are highly desirable to avoid loss of usefulness or efficiency of materials during the repair process. But examples that show rapid healing are rarely reported in the literature. Considering that the sample length is in the range of millimeters, with an ultrafast motion of the fragments and exceptional fast-healing, we used a high-speed camera (with a frame rate of ~1200-1600 fps) equipped with a stereomicroscope and recorded the fracture-healing events in detail. The healing time is assessed by measuring the time lapse between the frame (video-grab) in which the withdrawal of the force on the fractured crystals is noticed and the frame with complete disappearance of the crack-line between the two recombined shards. Using a series of single crystals, we determined that the healing period

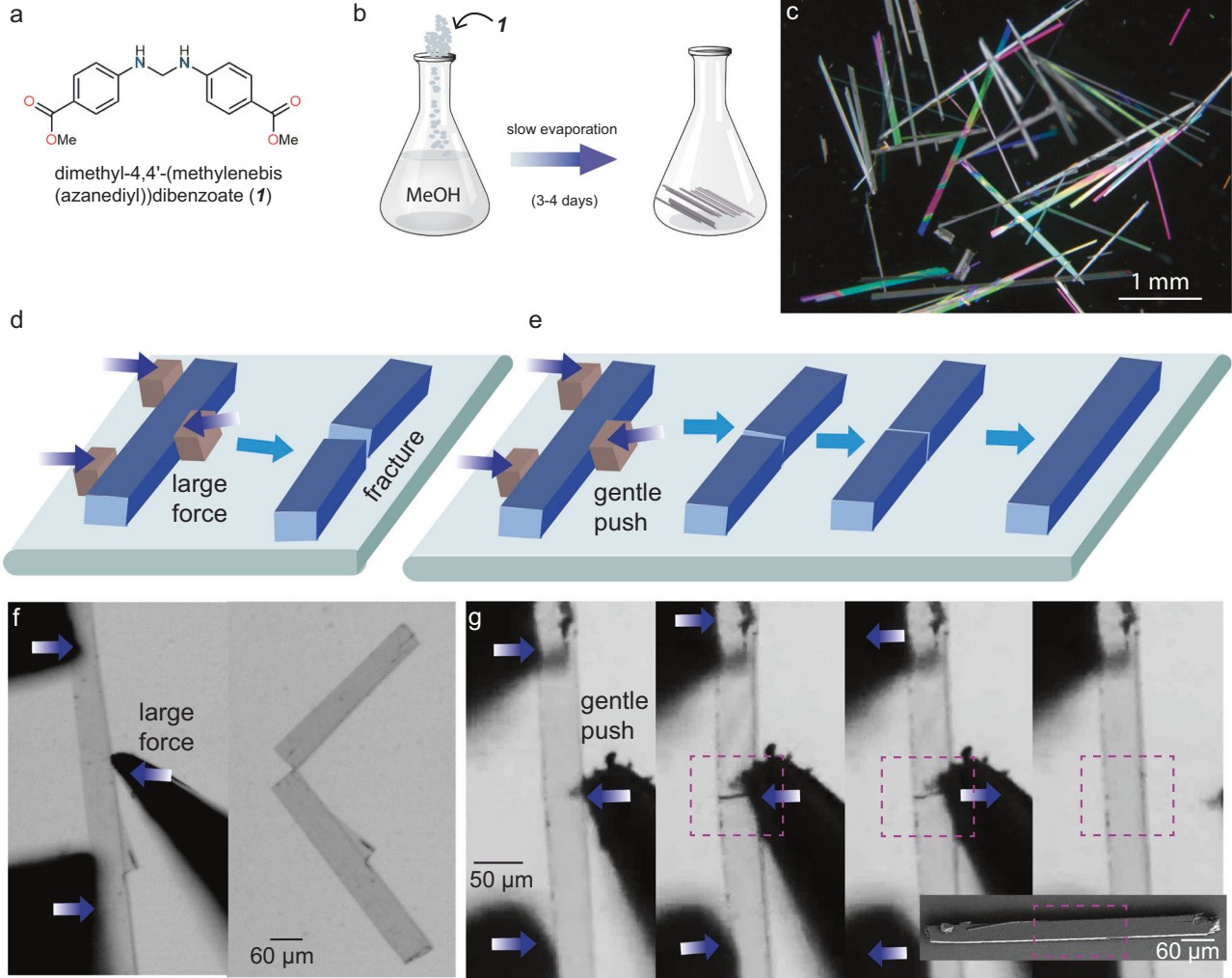

**Fig. 1 | Demonstration of autonomous self-healing in single crystals of 1.**
**a** Molecular structure of **1**. **b** Slow evaporation crystallization method to obtain single crystals. **c** Dark field image showing acicular single crystals under a cross polarizer. Schematic representations of **d** brittle fracture of crystals under excessive mechanical stress, and **e** application of a gentle mechanical stress leading to fracture, followed by autonomous self-healing upon withdrawal of force. Optical image of a crystal showing **f** brittle fracture; and **g** fracture followed by autonomous self-healing upon withdrawal of force (bottom inset: SEM image of the same healed crystal with no crack-line).

in these crystals is in the range of 10-30 ms (Supplementary Fig. 5 and 6). Notably, this is one among the fastest autonomous repairs in any type of self-healing materials[1].

## Crystal structure

To establish the structural basis for the self-healing and mechanical actuation response of **1**, we determined its crystal structure and analyzed the packing. It crystallizes in the tetragonal non- centrosymmetric (piezoelectric) polar space group $I4_1cd$ with half molecule in the asymmetric unit. Due to the presence of a $sp^3$-hybridized, tetrahedral methylene connector ($-CH_2-$), the molecule adopts a V-shaped geometry, with an angle of 105.48° between the two arms. These V-shaped molecules, which sit on a two-fold axis, close pack along $c$-axis with a criss-cross arrangement and form columns. Notably, the molecules are not involved in π-stacking interactions; instead, they interact via weak $C-H\cdots\pi$ ($d$[Å], $\theta$[°]: 2.92 Å, 166.20°). With the $-CH_2-$ groups of all the V-shaped molecules in columns pointed to the same side, the net dipole moment is expected to orient along the crystal needle axis, i.e. $c$-axis here (Fig. 3a). Each molecule contains two potential hydrogen bond donors, namely N−H groups at the mid-region and two hydrogen bond acceptors, namely -C=O groups of -COOMe functionalities, at either ends of the molecule. The molecules form parallel columns,

connected via strong N−H⋯O (2.03 Å, 163.82°) (amine-to-ester) and supportive C−H⋯O (2.49 Å, 140.15°) interactions. These hydrogen bonds connect the molecules in the three dimensions (3D) (Fig. 3b). Besides, the structure has dispersive interaction zones in the crystal packing. The overall density of hydrogen bonds in the structure is moderate with interaction strength comparable in three dimensions (Fig. 3c, d). In our earlier study, we reported that the mechanical fracture in some non-centrosymmetric crystals can generate surface charges[1]. Exact structural origins as to why this occurs only in some of the non-centrosymmetric crystals and not in all, is yet to be fully understood.

## Nanoindentation and energy-frameworks analysis

We employed the nanoindentation technique[50] to characterize the nanomechanical response of crystals of **1**. We obtained several load-depth ($P$-$h$) curves by indenting on side (100)/(010) and top (001) faces of the crystals (Fig. 3e). The extracted elastic modulus, $E$, and hardness, $H$, (7.50 ± 0.15 GPa and 0.60 ± 0.02 GPa, respectively) for (100)/(010) and (7.1 ± 0.3 GPa and 0.50 ± 0.04 GPa, respectively) for (001) show that all the faces have comparable mechanical properties. The intra ($C-H\cdots\pi$ and dispersive) and inter-column (N−H⋯O and C−H⋯O) intermolecular interaction energies of −43 kJ/mol and

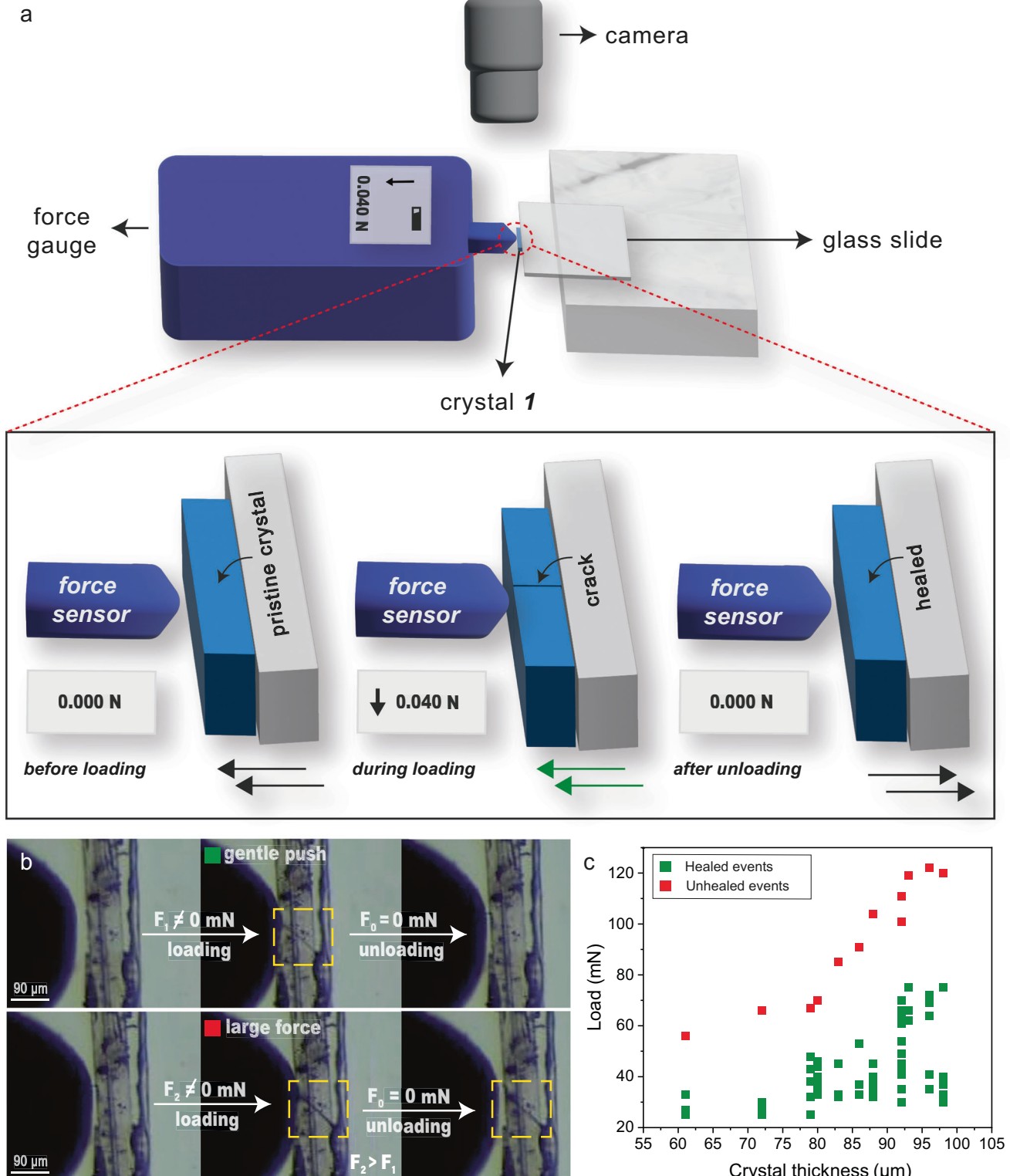

**Fig. 2 | Quantification of mechanical forces required for self-healing.** **a** Schematic representation of mechanical loading performed on the crystals of *1* using a force sensor (arrows towards left and right indicate the translation direction of the stage; the green arrow represents a gentle force, which is within the limit for observing crystals' self-healing). **b** Optical image of a crystal under gentle push (load 40 mN; upper panels) and large force (load 74 mN; lower panels). **c** Load vs crystal thickness plot from multiple crystals in loading cycles (green and red symbols correspond to healed and unhealed events under gentle and excessive force, respectively).

−37 kJ/mol, respectively, (Fig. 3c, d) are comparable, and so are the stresses required to deform the crystal on the two faces under the indenter tip. Besides, the pile-up after the indent impressions indicates the incompressible nature of the crystals.

## Ultrafast autonomous actuation

Although the research on stimuli responsive molecular crystals gained attention only in the recent years, numerous single crystals have been reported that display bending[15,17], jumping[51], twisting[19,52], rolling[53],

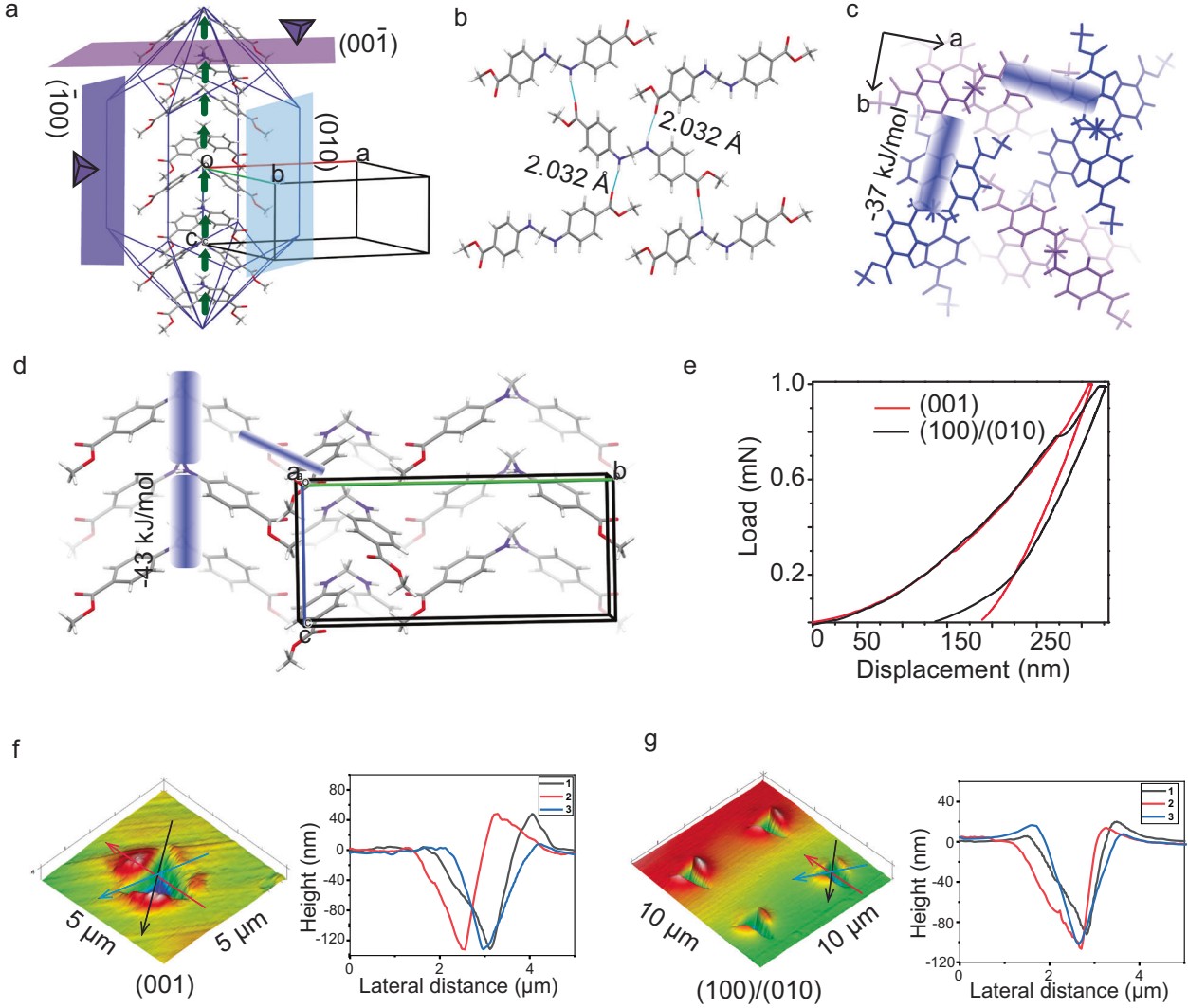

**Fig. 3 | Structure-mechanical property correlation in crystals of *1*. a** Calculated BFDH crystal morphology is shown along with the orientation of molecular dipoles and crystal faces used for nanoindentation experiments. **b** Crystal packing views shown with hydrogen bonding interactions and **c, d** pairwise interaction energies. **e** Representative load-depth curves obtained from nanoindentation on (001) and (100)/(010) faces. **f, g** Scanning probe microscope (SPM) images of indent impressions with corresponding height profiles obtained from nanoindentation on (001) and (100)/(010) faces, respectively.

splitting[54], self-healing[2], shape memory[55], etc. On the other hand, though thermosalient crystals are known for their one-stroke high work output[18], the resultant motions generally introduce infinite degrees of freedom. Further they always need some form of external stimulus for the aforementioned actions to happen. A mechanical actuator, in theory, is a material (or device) that does mechanical work in response to another input. An actuating element made of responsive, dynamic material is the main work-producing component within a machine. Direction-dependent motion with appropriate work output/stroke/acceleration is an essential criterion for an actuator[56–58], but difficult to achieve in synthetic materials. Achieving such mechanical motion in a specified direction necessitates an adequate design strategy and the proper guidance of external stimuli (heat, light, electric/magnetic field etc).

The mechanical actuation behavior (Supplementary Movies 9–13) observed in single crystals of *1* is one of its kind and is distinct from the other well-known stimuli responsive single crystals reported so far. The crystals of *1* when broken into pieces by an external mechanical input, generate a significant amount of attractive force[1]. Mechanical actuation caused by attractive forces in piezoelectric single crystals has not been studied in detail till date. It is worth noting that pristine,

unperturbed crystals do not show any form of motion or notable attraction among themselves, but only when broken by an external force, they show such actuation motion. Therefore, in order to access some performance attributes of this unique single crystal, the ultrafast dynamics are studied systematically through two types of mechanical motions, determined by the pre-alignment of the shards, namely, (i) angular actuation motion and (ii) linear actuation motion.

(i) When a needle-shaped crystal is fractured in a three-point bending mode with a gentle stress, a linear crack propagates with a V-shaped geometry between two fracture surfaces, and when the external stress is released, coulombic-type force triggers angular actuation motion, helping to close the two surfaces, often resulting in a perfect self-healing. In this actuation mode, the fragments do not completely disintegrate, but rather stay in contact at the pivoted end of the crack (Fig. 4a–d). Upon removal of the load, generally, the lighter fragment accelerates towards its counterpart over an arc in an angular fashion, eventually uniting with it.

Note that, the morphology of the actuating pieces is assumed to be a rectangular block, and the radius is defined as the distance between the pivot point and the moving fragment's center of

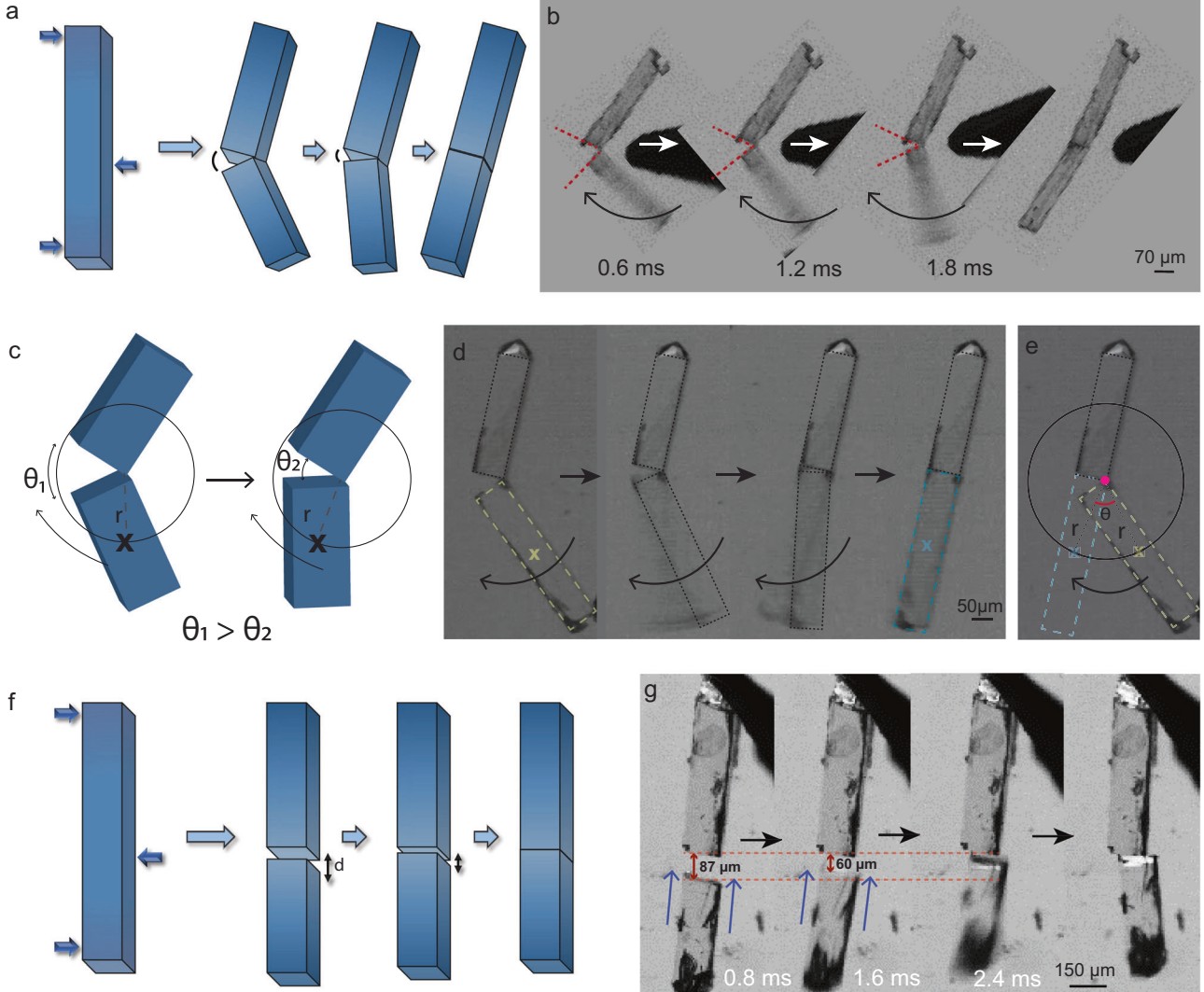

**Fig. 4 | Autonomous mechanical actuation. a** Schematic representation of angular actuation motion in the fractured crystals. **b, d** Optical images showing angular motion upon fracture of the crystals captured under a high-speed camera. **c** The crack closure mechanism described using an angular model where $\theta_1 - \theta_2$ is the angular displacement. **e** Angular motion traced by the center of mass of the actuating crystal of radius, $r$, and angular displacement, $\theta$. **f** Schematic representation of linear actuation motion in the fractured crystals separated over a distance, $d$. **g** Linear motion of fractured crystals, captured under a high-speed camera where the distance closes as the actuating piece moves nearer.

mass. We consider a circle of radius, $r$, with the pivot point as the circle's center by extending the locus of the center of mass (of the actuating shard) (Fig. 4c, d, e). The mechanical model presented in Fig. 4c, e and Supplementary Fig. 7 allowed us to calculate certain accessible performance attributes, which include average angular acceleration ($13 \times 10^3 - 71 \times 10^4$ rads$^{-2}$), torque ($0.15 \times 10^{-12} - 96 \times 10^{-12}$ kgm$^2$s$^{-2}$), and work capacity ($0.2 - 15$ Nm$^{-2}$) which are tabulated in Supplementary Table 2. The impressive numbers validate the ultrafast actuation behavior of the self-healing single crystals (Supplementary Movies 9 and 10).

(ii) To understand the linear actuation motion, we fractured the single crystals into two pieces and placed them collinearly at an optimal distance. The complementary surface charges that are present on both the fracture tips result in an attraction that pulls them together (Fig. 4f, g and Supplementary Movies 11-13). Evaluation of this motion (Fig. 4f, g and Supplementary Fig. 8) revealed impressive values of average linear acceleration ($1 - 62$ ms$^{-2}$), force ($3 \times 10^{-6} - 3 \times 10^{-4}$ mN) and work capacity ($0.04 - 2$ Nm$^{-2}$) (Supplementary Table 2).

Another prominent attribute which is necessary for any actuator is its actuation response time, which indicates how quick the actuation phenomenon is for a given application. The actuation observed in these crystals is in millisecond range ($0.6 - 8$ ms) (Supplementary Table 2) which is impressive and places this example above the most fastest stimuli responsive single crystals (Supplementary Fig. 9)[59].

## Crystallographic precision self-healing

Comparison of the diffraction profiles of the pristine and healed (with no visible crack-line after healing) single crystals reveal an excellent retention of the long-range order and crystallinity in the latter (Supplementary Fig. 10). Further, the indistinguishable 2D projections of reconstructed reciprocal images of pristine and completely healed crystals indicate an efficient macroscopic domain-alignment and neat-healing at the level of the crystallographic detection limit in the latter (Supplementary Fig. 10a, b). Hence, these observations suggest that for all practical purposes, the self-healed crystals are single-crystalline. On the other hand, in the imperfectly healed crystal the two crystal pieces are misaligned at the crack-junction. Since each misaligned

crystal piece acts as a separate domain, the crystal produces split diffraction spots (Supplementary Fig. 10d).

## Self-healing efficiency by second harmonic generation

Nonlinear optical properties of materials such as second and higher order harmonic generation, nonlinear refraction, and two photon absorption play salient roles in frequency conversion, ultrashort optical pulse generation, and so forth[43]. SHG occurs when a crystal absorbs two photons having the same energy simultaneously to excite any virtual state and relaxation happens with a new photon having the energy twice that of the initial photons[60–63]. The nonlinear optical properties are dependent on the crystal symmetry[37], transition dipole moments[40], specific optical excitation[40], and local environment of molecules[38]. Hence, restoration of crystal's integrity through self-healing can be examined through SHG measurements. We excited the crystal $1$ by the laser having a peak wavelength nearly at 810 nm and obtained a spectrum of the emergent light peaked at 405 nm, which confirmed its ability to produce SHG (Supplementary Fig. 12b). As SHG is a second order nonlinear response and involves two photons for excitation, the intensity of the emitted SHG light should vary as the

square of the incident light intensity. Hence, we further studied the power dependence of the SHG response. The slope of the SHG intensity vs. excitation power in the log-log scale is found to be $2.01 \pm 0.01$, confirming that the emergent light is due to the SHG response of the crystal $1$ (Supplementary Fig. 12c). We also analyzed the full-width-at-half-maximum (FWHM) of the spectra of both the SHG signal and the incident laser beam. Theoretical analysis predicts the ratio of FWHM of the SHG spectrum to that of the incident laser spectrum to be $0.354$[64], which is consistent with our experimentally observed value of $0.36 \pm 0.01$. This further confirmed the SHG activity of crystal $1$.

We further performed the polarization dependent SHG study of crystal $1$ to investigate the response from its anisotropic structure. Considering the beam's reference frame $(x, y, z)$ to be parallel to the crystal frame $(a, b, c)$, an incident beam propagating along $x$-axis, as shown in Fig. 5a, was focused on the crystal surface, (100) face. It is noteworthy that the crystal system being tetragonal, the angle among the crystal axes is 90°. Initially, the laser beam was linearly polarized along the $y$-axis. Then the linear polarization direction of the incident laser beam was rotated from the $y$-axis towards the $z$-axis by rotating a half-wave plate in the beam path. The intensity of the SHG signal was

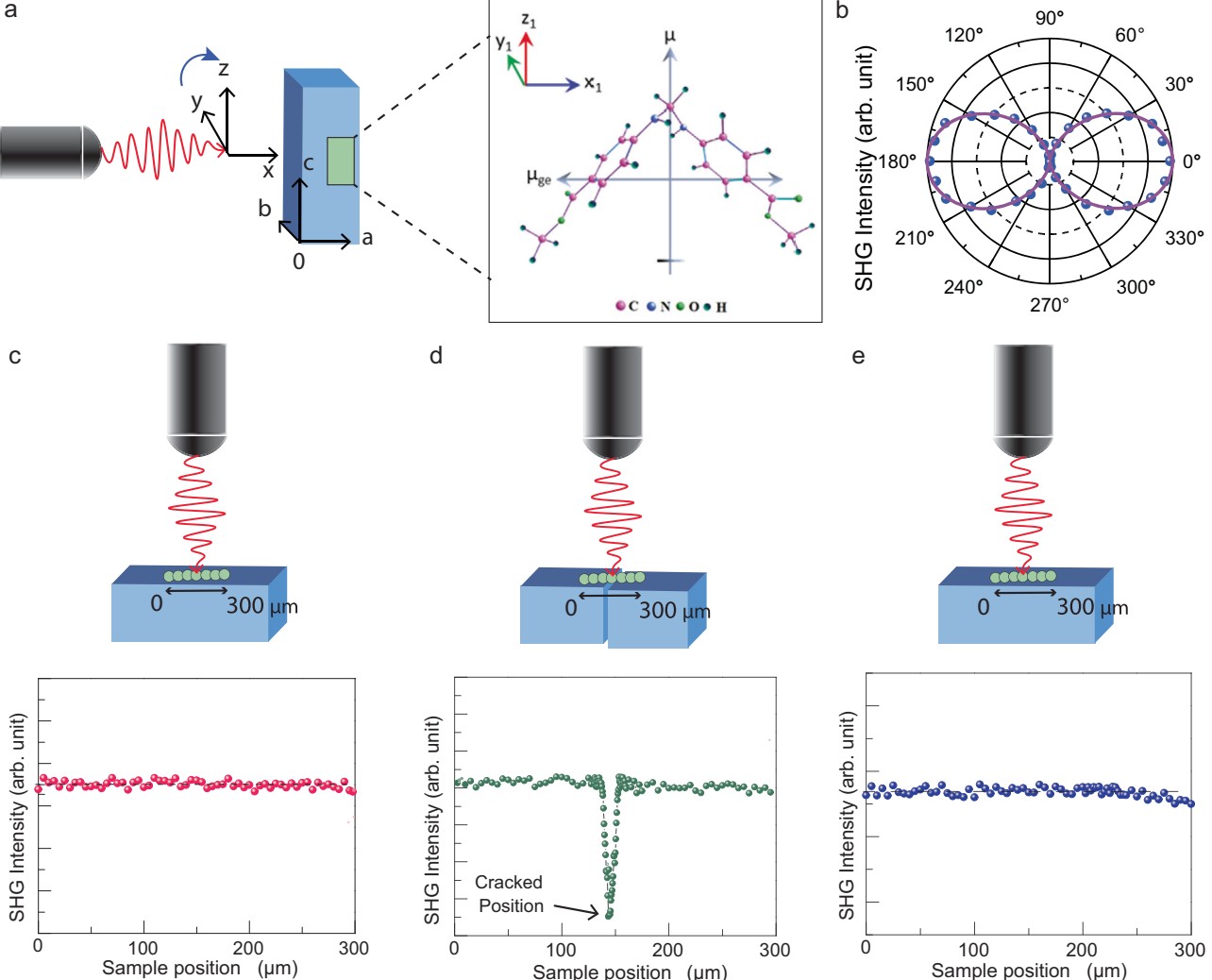

**Fig. 5 | Demonstration of self-healing in crystals of $1$ using SHG studies.**
**a** Incident laser beam of 810 nm illuminated on (100) face of the crystal and different dipolar orientations of a molecule when viewed perpendicular to (010) face. **b** SHG polar plot obtained by rotating the linear polarization direction of the incident laser from $y$ to $z$ direction. **c**–**e** $y$-polarized laser beam incident on (100) face of the pristine, imperfectly healed and healed crystals, respectively. A series of scanning, line-mapping SHG measurements along $c$-axis of the crystals indicate that the SHG intensity suddenly drops in the vicinity of crack line in case of an imperfectly healed crystal while in both the pristine and self-healed crystals, the SHG intensity remains almost the same (or uniform) throughout the scan region.

recorded for a full 360° rotation of the linear polarization direction of the incident laser beam. From the polarization-dependent SHG experimental results, as shown in the Fig. 5b, it is clear that the SHG response of the crystal is maximum when the electric field of the incident beam is perpendicular to the needle axis. By fitting the polarization dependent SHG response, we determined the ratio of the second-order nonlinear optical coefficients[37] $\frac{d_{15}}{d_{33}} \approx 3 \pm 0.02$ and $\frac{d_{31}}{d_{33}} \approx 7 \pm 0.11$ (see Methods). The ratio depends on the molecular orientation in the crystal packing. Due to the presence of electron-donor and electron-acceptor substituents, amino and carbonyl groups, respectively, a strong charge transfer interaction is expected to induce polarization in the system. Note that the SHG activity generally depends on polarization property. Furthermore, the findings are supported by the density functional theory (DFT) calculations on permanent and transition dipole-moment. The ground-state dipole moment, $\mu_g = 1.11$ Debye and excited-state dipole moment, $\mu_e = 2.11$ Debye of the molecules are aligned along the $c$-axis of the crystal, whereas the transition dipole-moment $\mu_{ge}$ (molecular $x_1$-axis) = 5.87 Debye and $\mu_{ge}$ (molecular $y_1$-axis) = 2.84 Debye of the molecules are oriented towards crystallographic $a/b$-axis (see Fig. 5a and Methods), which is in agreement with our experimental results of the ratio of second-order nonlinear coefficients.

Having characterized the SHG phenomena, we studied the pristine, imperfectly healed and healed crystals to explore the practical utility of the self-healing in molecular crystals. To begin with, the (100) face of the pristine crystals was illuminated with 810 nm laser beam of a focused spot (radius of ~ 8 μm). Keeping the laser polarization along $y$-axis, as SHG is maximum for this polarization, we translated the crystal parallel to $c$-axis with a step-size of 1 μm using motorized translational stage having 0.1 μm resolution. As shown in Fig. 5c, SHG line profile signal on a pristine crystal is nearly uniform along the length. Similar experiments performed on the healed crystal (Supplementary Fig. 2e, f and Supplementary Movie 4) confirmed that the SHG output remains nearly the same throughout the crystal, including at the vicinity of the healed junction (Fig. 5e). Whereas, for a imperfectly healed crystal (with a visible crack line) (Supplementary Fig. 3a, b), a sharp drop in the SHG signal at the crack-line was observed which reveals that the broken pieces could not rejoin perfectly in this case (Fig. 5d). Therefore, this infers that healed crystals are as good as the pristine samples with respect to SHG activity, demonstrating the excellent technological importance of self-healing crystals.

## Discussion

Here, we report a dibenzoate single crystal that is stable in moderate-to-harsh environmental conditions, and which upon gentle stress, is capable of autonomous self-healing in a few milliseconds without needing any external intervention. The crystals of *1* fracture in a brittle manner under excessive mechanical stress, generally leading to imperfect healing. Provided the fragments lie within the operational distance, they attract through complimentary surface charges and recombine. Our quantitative analysis of the mechanical loads revealed that the self-healing can be observed in repeating cycles under the gentle external loads (typically the loads being <70 mN). Self-healing efficiency was studied using SCXRD, which demonstrated retention of structural integrity after healing. As a proof-of-concept, SHG studies were conducted on pristine, imperfectly healed and healed crystals wherein the SHG output remained the same for the pristine and neatly healed samples, confirming the excellent self-healing efficiency. It is worth noting that the crystals rejoin even when the fractured pieces are several tens of micrometers apart via an ultrafast automated actuation motion. However, perfect healing was not observed due to misalignment. This unique, spontaneous actuation is automatically triggered with an average response time of these crystals being faster than most of the reported stimuli-responsive single crystals. We also statistically calculated the performance metrics of these single crystals

using two actuation motions (linear and angular) that can be applied to other molecular crystals that develop charges. Numerically, the fractured crystal shards actuate with accelerations comparable to a light motor vehicle. Despite their small size, their work capacity falls in the range of MEMS (micro-electromechanical systems) and outperform electroactive polymers.

In summary, our organic self-healing crystals not only can efficiently heal microcracks but also self-heal even when fragmented, which is remarkable although it is with low propensity. Hence, self-healing properties incorporated in single crystals will make them much more durable and vastly reduce maintenance costs overall in many practical applications such as piezoelectric actuators, ferroelectric memory devices, SHG frequency modulating devices, and so on, and may open up multiple opportunities in materials science, optics, and soft robotics.

## Methods
### Synthesis and crystallization
Dimethyl-4,4'-(methylenebis(azanediyl))dibenzoate was synthesized using the following method[65]. To a stirred solution of methyl 4-aminobenzoate (1.51 g, 10 mmol) in 15 mL acetonitrile, 1.9 mmol (81.96 μL) of 98% formic acid was added. This solution was then mixed with the previously prepared solution of 5 mmol (376.5 μL) of 37% formaldehyde in 5 mL acetonitrile, which was then stirred for 2-3 hrs to obtain a precipitate. The precipitate was filtered out and washed with 30 mL acetonitrile followed by 30 mL hexane to get pure product (~ 1 g) with a yield of nearly 60%.

Reagents (methyl 4-aminobenzoate, formic acid, formaldehyde) were purchased from Sigma Aldrich company and used without further purification.

$^1$H NMR (DMSO-$d_6$), δ):7.66 (d, 4H), 7.24 (t, 2H), 6.69 (d, 4H), 4.54 (t, 2H), 3.70 (s, 6H).

HRMS (ESI) $m/z$: [M + Na]$^+$ calcd for $C_{17}H_{18}N_2O_4$, 337.1164; found, 337.1159.

After characterization of the compound with NMR and HRMS, 25 mg of the product was added to 5 mL of methanol in a clean, dust free conical flask, and heated for 25 minutes at ~ 90 °C on a hot plate to dissolve the compound. After that the solution was kept for slow evaporation for about 3-5 days to obtain clean, needle shaped single crystals of *1*.

### Single crystal X-ray diffraction (SCXRD)
X-ray diffraction data for crystal *1* was obtained at 293 K using a Rigaku (dual, Cu/Mo at zero, Eos) diffractometer with monochromatic CuKα (λ = 1.54184 Å) source having a 100 μm beam size. The structure was solved using Olex2, (1.2.9 version)[66] using the structure solution program, SHELXT and refined with the refinement package, SHELXL[67], using the Least Squares minimization approach. The anisotropic displacement parameters of all non-hydrogen atoms were refined. Mercury (3.10.1 version) was used to create all of the crystal packing diagrams.

### Powder XRD
The PXRD patterns were obtained on a Rigaku Miniflex with a Cu-Kα radiation. The tube voltage and amperage were set at 40 kV and 50 mA, respectively. Sample was scanned between 5° and 50° 2θ with a step size of 0.02°. The instrument had previously been calibrated using a silicon standard[1].

### Scanning electron microscopy (SEM)
The images were produced with the help of Zeiss DSM 950 and FEI QUANTA 200 3D microscope operating at 10 kV with tungsten filament serving as the electron S4 source. An SCD 040 Balzers Union sputterer was used to sputter gold (nano-sized film) onto the samples prior to imaging in order to prevent charging during SEM analysis[1].

## Kelvin probe force microscopy (KPFM)

The surface topography and nanoscale surface potential were measured, using the noncontact KPFM technique, with a 20 minutes lapse from the time of fracture. Cypher ES equipped with ASYELEC-01-R2 AFM tip, coated with Ti/Ir (tip radius of 25 nm), resonance frequency of 75 kHz, and force constant k of 2.8 N m$^{-1}$ were used to conduct the experiment. KPFM scan was performed with a lift height, d = 50 nm and $V_{AC}$ = 1 V. KPFM has a spatial resolution of around 25 nm. The topographic scan was performed by using tapping mode. The work function of the tip used was ~ 4.75 eV.

## Nanoindentation

Nanoindentation experiments were performed on the (100)/(010) and (001) faces of single crystals of **1** at maximum load of 1 mN using the TI Premier from Hysitron, Minneapolis, USA, equipped with an in-situ Scanning Probe Microscope (SPM). To determine the hardness (H) and elastic modulus (E) of the crystals a Berkovich tip (three-sided pyramidal tip with a total included plane-edge angle of 142.3°) of radius ~150 nm was used[1]. The standard Oliver-Pharr (O&P) method[68] is used to extract the H and E.

Please note that we did not exceed 1 mN load in the case of the samples of **1** as it was challenging to clean the indentation tip, which routinely became dirty owing to crystal debris which are not readily soluble in the majority of organic solvents at ambient conditions.

## Thermal gravimetric analysis (TGA)

TGA experiments were conducted on a TG50 analyzer (MettlerToledo) and an SDT Q600 TG-DTA analyzer under N$_2$ atmosphere at a heating rate of 2 °C min$^{-1}$ within a temperature range of 25 °C to 275 °C.

## Differential scanning calorimetry (DSC)

DSC experiments were performed on a Mettler-Toledo DSI1 STAR$^e$ instrument on accurately weighed samples (4 mg) placed in hermetically sealed aluminum crucibles (40 µL) upon scanning in the range of 25 °C to 275 °C at a heating rate of 5 °C/min under a dry nitrogen atmosphere (flow rate 80 mL/min)[1].

## Second harmonic generation (SHG)

The schematic of the setup has been shown in the Supplementary Fig. 11. Excitation beam from a Ti:sapphire laser (80 MHz repetition rate, 100 fs pulse width) having 100 mW of power has been focused (spot radius ~8 $\mu m$) on to crystal **1**. Second harmonic signal collected by the collection lens enters into the spectrograph after reflected from two dichroic mirrors which eliminates the transmitted excitation beam from entering into the spectrograph. The spectra of the SHG signal were measured by the CCD attached with the spectrograph. Polarization-dependent SHG measurements have been performed by rotating a half-waveplate placed in the path of the excitation beam.

Translating the sample by a motorized stage with sub-micrometer precision, we have measured the SHG response from different positions of the crystal. Position-dependent measurements have been performed for the polarization angle for which the SHG response is maximum.

At the high-intensity limit of SHG response, the induced second-order polarization (P) having the components $P_x$, $P_y$ and $P_z$ has a specific relationship with the different electric field components of the polarized light. This can be described by the following equation 1 for point group 4 mm after applying the intrinsic permutation symmetry and spatial symmetry for uniaxial crystal. Using the symmetry operation one can write $d_{24} = d_{15}$ and $d_{32} = d_{31}$ and the dependence of the second-order polarization components (responsible for SHG) on the electric field components can be expressed as

$$\begin{bmatrix} P_x \\ P_y \\ P_z \end{bmatrix} = 2\epsilon_0 \begin{bmatrix} 0 & 0 & 0 & 0 & d_{15} & 0 \\ 0 & 0 & 0 & d_{15} & 0 & 0 \\ d_{31} & d_{31} & d_{33} & 0 & 0 & 0 \end{bmatrix} \begin{bmatrix} E_x^2 \\ E_y^2 \\ E_z^2 \\ 2E_yE_z \\ 2E_zE_x \\ 2E_xE_y \end{bmatrix} \tag{1}$$

where, $d_{ij}$ is the second-order optical susceptibility tensor for the crystal under observation, and $E_x$, $E_y$, and $E_z$ represent the components of the electric field of the incident light. The detected intensity of the SHG signal is proportional to the square of the induced second order polarization (P). As the light propagates along the x-axis (a-axis of the crystal), we have $E_x = 0$. The crystal needle-axis (c-axis) is along the z-axis. Thus the intensity of the SHG can be expressed as

$$I_{SHG} \propto P_x^2 + P_y^2 + P_z^2 \tag{2}$$

$$I_{SHG} = 4\epsilon_0^2 \left[ 4d_{15}^2 E_y^2 E_z^2 + \left( d_{31}E_y^2 + d_{33}E_z^2 \right)^2 \right] \tag{3}$$

Now in our case $E_y = E \cos\theta$ and $E_z = E \sin\theta$. Using these electric field components one can rewrite the eq. 3 as,

$$I_{SHG} = A \left[ d_{15}^2 \sin^2 2\theta + \left( d_{31}\cos^2\theta + d_{33}\sin^2\theta \right)^2 \right] \tag{4}$$

where, $A = 4\epsilon_0^2 I_{ex}^2$; $I_{ex}$ = excitation laser intensity $\propto E^2$

From the experimental observation of the polarization-dependent SHG response (as shown in the Fig. 5a) it is clear that the SHG response of the crystal is maximum when the electric field is perpendicular to the needle axis. We have fitted the dependence of SHG intensity on the input light polarization to equation 4. We have calculated from the best fit parameters,

$$\frac{d_{15}}{d_{33}} \approx 3 \pm 0.02 \text{ and } \frac{d_{31}}{d_{33}} \approx 7 \pm 0.11$$

We have measured the polarization dependence of the SHG intensity both for the neatly healed position and fractured (imperfectly healed) position, respectively, for healed and imperfectly healed crystals (see Supplementary Fig. 13). In both the cases, the variation of the SHG intensity with the polarization direction of the input light beam is very similar. However, the overall SHG intensity is significantly reduced for the imperfectly healed crystal. Equation 4 fits well with the polarization-dependence of the SHG intensity for both the imperfectly healed and healed crystals and the ratios of the d-coefficients obtained from the fits remain same as those for the pristine crystals. The reduction in the SHG intensity for the imperfectly healed crystal is on account of the reduction of the crystalline substance at the fracture-line (imperfectly healed) position of the crystal.

## Mechanical manipulation under a high-speed camera integrated with an optical microscopy

The single crystals were mechanically manipulated using a pair of forceps and a needle head under a Leica (M205 FCA) polarized optical microscope with a Fastcam Mini high-speed camera[1]. The magnification employed was between 8 to 12 times, and the movies were captured at 1250 and 1600 frames per second (fps) at 1280 × 1024 resolution.

Fastcam software was used to analyze the videos, to observe the mechanical healing and actuation in slow motion and real time.

## Mechanical manipulation under an optical microscopy attached with a digital camera

Mechanical manipulations of the single crystals for multiple healing events and quantification experiments were performed under an optical microscope with a Leica camera. The images and movies were captured at real time.

## Energy frameworks calculations

The calculations pertaining to intermolecular interactions were performed using the software suite Crystal-Explorer17 based on Gaussian B3LYP-D2/6-31 G (d, p) molecular wave functions calculated using CIF files[69].

## Density functional theory (DFT) calculations

The geometry optimization has been carried out by taking the crystal structure as a reference geometry. The hydrogens are optimized by freezing all heavier atoms to retain the crystal arrangement. The optimizations are carried out by using B3LYP functional and 6-31G(d) basis set. The electric-dipole properties of the molecule were calculated at the same level of theory. The excited-state and transition dipole moments are calculated by using the time-dependent DFT (TDDFT)//CAM-B3LYP//6-31G(d) method. All calculations are performed by using Gaussian 16 (G16) program[70].

Molecular dipole moments: The polar two-fold molecular axis is equivalent to the $z_1$-axis and is parallel to the $c$-axis of the crystal arrangement, molecular $x_1$-axis and $y_1$-axis are parallel to the $a$-axis and $b$-axis of the unit cell, respectively for the molecule shown in Fig. 5a. As the crystal is tetragonal the $a$ and $b$- axes are equivalent and both are unparallel with the c-axis, there is another set of molecules where molecular axes $x_1$ and $y_1$ become parallel to $b$-axis and $a$-axis of the unit cell, respectively. The ground state dipole moment of the molecule is 1.11 Debye ($\mu_0 = 3.68 \times 10^{-30}$ C m) and it is oriented along $z_1$ -axis, as shown in Fig. 5a. However, the second harmonic generation (SHG) is highly dependent on the difference between the permanent dipole moment of ground state and excited state ($\Delta\mu = \mu_e - \mu_g$) and the transition dipole moment ($\mu_{ge}$)[40]. The permanent dipole moment corresponding to the excited state is 2.11 Debye which estimates the $\Delta\mu$ as 1.00 Debye. $\Delta\mu$ is also oriented along molecular $z_1$-axis. The lowest optically active excited state corresponds to 4.92 eV and the excitation mainly involved the transition between HOMO and LUMO (70%) with the transition dipole moment along molecular $x_1$-axis.

## Quantitative analysis of mechanical forces for healed and unhealed events

To quantify the load responsible for self-healing, crystals of **1** of different thicknesses were attached to edge of a glass slide using small amount of glue. Then the stage having the crystals was translated towards the tip of a force sensor (accuracy of ±1 %) with a minimum count of 1 mN having tip diameter of ~190 μm. The loads involved in gentle pushes for observing different healing cycles and the loads involved in excessive forces causing imperfect healing or fracture in crystals were measured and analyzed. Please note that the load values obtained from the instrument are directly used here to plot. Due to the absence of standardized methods to extract precise contact area from the new experimental tool we used, we have provided the raw load values only (to avoid errors in calculating the threshold pressure values).

## Healing time

The healing time (the time between the releases of mechanical force after mechanical crack generation till the disappearance of the crack)

was measured for different crystals mainly at 1250 or 1600 fps (see Supplementary Fig. 5 and 6).

## Performance parameters

The actuating crystal pieces' linear displacement and angular displacement were measured using P.F.V. (*Photron Fastcam Viewer*), and the motion was recorded under 1250 or 1600 fps using *FASTCAM Mini UX50*.

(i)   Response time. The duration of actuation is regarded as the actuating material's response time.

(ii)  Average linear acceleration. It is the average linear velocity (of the actuating crystal shard) divided by its actuation time.

(iii) Force output. Newton's second law $F = ma$ gives the average force where; $F$ is the net force (we exclude friction here), $m$ is the actuating mass, and $a$ is its average acceleration.

(iv)  Average angular acceleration. It is the average angular velocity (of the actuating crystal shard) divided by its actuation time. The radius here is the length taken from the pivot point to the center of mass of actuating body.

(v)   Moment of Inertia. It is taken about the axis $Z$ perpendicular to the plane of the pivot point. We approximated the crystal morphology ($l \times \omega \times t$) to be a rectangular block.

$$\text{Moment of Inertia about Z axis is given by } I = \frac{m}{3}(\omega^2 + l^2) \quad (5)$$

(vi)  Torque. It is the rotational equivalent of linear force, which causes an object to rotate about the axis $Z$ (in above case). It is the product of the moment of inertia (of the actuating crystal shard) about the axis $Z$ and the average angular acceleration associated with it.

(vii) Work capacity. The ability to do work is one of the essential attributes for assessing these materials' potential in actuation devices and systems. Work capacity is the maximum work done per unit volume.

## Data availability

All data are available from the authors on request. Crystallographic data for structures reported in this article have been deposited at The Cambridge Crystallographic Data Centre (CCDC), under deposition numbers 2183527 and 2184474. These data can be obtained free of charge from www.ccdc.cam.ac.uk/data_request/cif. A source file containing the coordinated of the optimized structures is present.

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

## Acknowledgements
We are thankful to Prof. Ananda Dasgupta and Prof. Soumitro Banerjee from Department of Physical Sciences, IISER Kolkata for their valuable suggestions in calculating the actuation parameters. S.M. thanks CSIR India for fellowship (File No. 09/921(0183)/2017-EMR-I). S.R. and S.B. thank the DST-INSPIRE fellowship. R.C. thanks KVPY (SX-1411075). C.M.R. thanks DST-SERB for funding (No: CRG/2021/004992). A.K.P. thanks Technical Research Centre (TRC) in IACS for Research Associate fellowship. A.D. thanks SERB for funding (CRG/2020/000301).

## Author contributions
S.M. carried out all mechanical manipulations of crystals. R.C. and S.M. synthesized the molecule. S.M. collected SCXRD and SEM data. P.T. quantified the performance parameters and response time, and analyzed healing timescale with S.M. The SHG studies were carried out by S.R., and S.R., B.P. analyzed the SHG data. S.M. performed nanoindentation experiments with the help of S.B. DFT calculations were carried out by A.K.P. and the analysis of the output results analyzed by A.K.P. and A.D. S.M., S.B. and C.M.R. analyzed the crystal structure. The experiments to quantify loads for self-healing events were performed by S.M. and P.T. and analyzed the results. S.M. and C.M.R. planned the experiments for quantification of loads for self-healing and repeatable self-healing cycles. S.M., S.B. and C.M.R. planned all other experiments, analyzed the results. S.M., S.B., P.T. and CMR co-wrote the manuscript with inputs from all authors.

## Competing interests
The authors declare no competing interests.
