## [Peer Review File · Nature Communications]

Autonomous self-healing organic crystals for nonlinear opticsReviewers' Comments:

Reviewer #1:

Remarks to the Author:

This is one other example by C. Malla Reddy's group of highly interesting piezoelectric organic molecular crystals with exceptional self-healing properties that were achieved via remarkably fast self-actuation. The material reported here, concerning self-healing time, outperforms the first self-repairing piezoelectric molecular crystal reported by the same group in *Science* in 2021 (ref. [1]). Moreover, the self-healed crystals (reported in this MS) demonstrate excellent efficacy of the second harmonic generation, which consequently presents a great potential for application in high-end technologies.

However, several major points need to be addressed before the acceptance of the manuscript.

- In the Introduction, it was claimed that 'The crystals do not degrade even after one year, insoluble in most of the organic solvents at ambient condition and have high thermal stability upto ~ 200 °C (Supplementary Fig. 1), making it a better self-healing material over the previously reported crystals1.' The data supporting the statement would be needed as these are not available in either of the materials, ref [1] and this MS.

- In the Crystal Structure paragraph, it was stated that molecules in 1 are 'connected via strong N–H...O and supportive C–H...O interactions' which 'form a 3D hydrogen-bonded network', and that '...the structure does not have any flat slip planes but has dispersive interaction zones in the crystal packing'. It is important to clarify the two remarks and to distinguish between the two contributions as this will help understand the structural background of this interesting and highly desired phenomenon. This is important especially as the first report on the autonomous self-repair of molecular crystals (ref [1]) claimed that crystals of piezoelectric materials that did not show any evidence of self-healing (the piezoelectric nature of a material 'is not a sufficient condition for self-healing') were comprising 'predominantly strong hydrogen-bonding networks and minimally dispersive interactions'.

Also, a clear envisioning/description of the 'criss-cross arrangement' as well as the depiction of the dispersive interaction zones, in addition to HBs (Fig.2), in the crystal packing of 1 would help understanding.

- The exceptional healing ability of cracks (1-3 μm in length) with appreciable healing propensity was claimed, and the authors refer to 'statistical data' of successfully healed crystals. It would be advisable to have the claim supported (in the main text) by numerical data (i.e. the percentage of the healed crystals; the success rate was actually 10-15%).

Also, Fig.4 in Supplementary does not present 'statistical data on fracture...', but video grabs of healed (2)/fractured (2)/imperfectly healed crystals (2) out of 20 tested crystals; the figure caption needs to be improved.

- It is not clear how the mechanical fracture tests on crystals of 1 showed that 'they develop long-lasting opposite charges on broken surfaces,...'. This needs clarification. No data on the surface potential of 1 was available nor it was reported that the experiments were performed either.

- To present the autonomous actuation, linear acceleration, force and work capacity were calculated from the volume and mass of the crystal portion, and the distance (and time!) the crystal portion travelled. All the data but time are presented in Table 8, Supplementary, which needs to be included too.

- Lns 123, 212; 'enormous' amount of attractive force; it is apparent that the restoring force between the two parts of the fractured crystals was strong indeed, but a term that describes the attraction more realistically would be required.

Also, some mostly technical issues need to be resolved as well:

- The 'Results' and 'Discussion' sections; the authors are invited to re-examine the entitling of the sections, especially the latter one which more closely corresponds to Conclusions.

- Capture to Fig.2e; for consistency reasons, the same face indexing should be used in the figure as in the capture to it; e.g. (100)/(0101) vs. (100). Please do correct it.

- The '>'-shaped' term; although it indeed follows the orientation of the fractured crystal in Figure 3, 'V-shaped' would be better suited and generally acceptable.

- Ln119, '... stereo-microscope equipped with high-speed camera(Fig. 1d-g).' Actually, Fig. 1c-d presents schematics. Please do correct it.
- Careful proofreading of the manuscript is required:
- e.g., Ln 144; '... we measured the time required for the cracks to disappear from the time of withdrawal of the load.', etc.
- Supplementary, Figs. 5 and 6; is there any need for separating the experiments into two figures and starting Fig. 6 with the label (e) instead of (a)?

Reviewer #2:

Remarks to the Author:

This is an interesting work that reports self-healing in molecular crystals. As an evidence of self healing, authors used SHG but claimed attractive forces at the fractured surface were responsible for self healing. This claim was not reconciled by experimental evidence. Authors also need to clarify what they meant by domain structure in diffraction of imperfectly healed crystals.

Reviewer #3:

Remarks to the Author:

- What are the noteworthy results?

The authors report on organic single crystals, which show the ability to heal ruptures over several μm in a fast manner. As claimed by the authors even in unprecedented speed. The structure of the crystals was established by DFT and XRD. The successful healing of crystal "1" was shown by XRD of a single self-healed crystal. The authors also show that optical properties (1 is SHG active material) are fully regained upon self-healing. Thermal stability of the organic crystals is shown by TGA and DSC measurements. Mechanical properties are measured by means of nanoindentation.

- Will the work be of significance to the field and related fields? How does it compare to the established literature? If the work is not original, please provide relevant references.

As also pointed out by the authors and shown in an extensive literature research, there is ample use for those properties in applications (autonomous actuators, optical applications). Here, the autonomous self-healing without any additional stimuli and the speed of shape recovery shown by means of a high-speed camera is emphasized and show soignificant progress in this field.

- Does the work support the conclusions and claims, or is additional evidence needed?

The study works as a proof of concept for claimed properties of crystal 1 and hence is in my opinion successful. The authors also show that the behavior of crystal 1 can be repeated in other crystals of the same batch.

- Are there any flaws in the data analysis, interpretation and conclusions? Do these prohibit publication or require revision?

At some points in this study, the authors are not clear on distinguishing a proof of concept from a statistical approach

1) p.5 line 131 to 134 the authors claim to provide statistical data in the Supplement Fig. 4. However, aforementioned Fig. 4 shows a series of images of successful and unsuccessful self-healings and the only mentioning of the success ratio: 2-3 of 20 crystals healed (Supplement line 49-53). This does not suffice as statistical treatment in my opinion.

2) In order to describe the phenomenon of the shown behavior of crystal 1 the authors make a very important distinction of self-healing behavior upon "excessive mechanical stress fracture in a brittle manner" (line 119-120) and "gentle force" (line 121). This is unfortunately not picked up in a hypothesis of the mechanism governing the self-healing.

Both points were not picked up in the Discussion (from line 341), which reads more as an outlook and does not critically discuss a hypothesis of the self-healing mechanism or the realistic use of this behavior for applications.

- Is the methodology sound? Does the work meet the expected standards in your field?

The authors explicitly state that the focus of this study is not on the mechanical properties of the crystals. And I applaud them for including nanomechanical measurements, which are not trivial with these materials. However, when presenting different mechanisms for material failure and plastic deformation (one brittle without self-healing option and one with fast self-healing option => line 119-121) it should be discussed how this influences mechanical measurements by nanoindentation. An instantaneous self-healing mechanism could make elastic and plastic deformations undistinguishable in the method of Oliver and Pharr. Hence, elasticity and hardness might be overestimated, which might jeopardize the author's claim that "this material is among the hardest and stiffest autonomous self-healing materials" (line 192).

- Is there enough detail provided in the methods for the work to be reproduced?

The study is well documented.

REVIEWER COMMENTS

Reviewer#1:

Comments: This is one other example by C. Malla Reddy's group of highly interesting piezoelectric organic molecular crystals with exceptional self-healing properties that were achieved via remarkably fast self-actuation. The material reported here, concerning self-healing time, outperforms the first self-repairing piezoelectric molecular crystal reported by the same group in *Science* in 2021 (ref. [1]). Moreover, the self-healed crystals (reported in this MS) demonstrate excellent efficacy of the second harmonic generation, which consequently presents a great potential for application in high-end technologies. However, several major points need to be addressed before the acceptance of the manuscript.

Reply: We thank the reviewer for the positive response and his/her encouraging statements on the importance of the autonomous self-healing crystals studied by second harmonic generation. We have addressed all the major comments, as follows, and we hope that the manuscript is now suitable for publication.

1. In the Introduction, it was claimed that 'The crystals do not degrade even after one year, insoluble in most of the organic solvents at ambient condition and have high thermal stability upto ~ 200 °C (Supplementary Fig. 1), making it a better self-healing material over the previously reported crystals¹.' The data supporting the statement would be needed as these are not available in either of the materials, ref [1] and this MS.

Reply: Thank you very much for raising these points.

As reported in the supporting information of ref [1], the bipyrazole crystals “become opaque at ambient conditions” with time and this may be due to gradual water loss, as also observed in its thermal analytic studies (TGA and DSC) (see below data).

From Reference 1:

Thermal analysis of the bipyrzole crystals of ref [1]. (A) TGA plot confirms stepwise water loss prior to melting. (B) The endotherms before melting in DSC confirms phase transitions due to the water loss.

Current study:

On the other hand, in the current study, our DSC and TGA data presented in the supplementary information (Supplementary Figure S1), it is evident that the crystals do not undergo any phase transition or show any noticeable mass loss until the melting temperature (see below data).

Supplementary Fig. 1: Thermal analysis of 1. a TGA plot of 1 (blue) and b corresponding endotherms in DSC (red). This confirms the superior thermal stability of our crystal (data is provided in the SI).

Again, in the present case of dibenzoate example, we observed that the crystals from all the batches remain transparent and stable for more than a year without any signs of degradation at ambient conditions (see below images).

Crystal 1 from all the batches (2.5 years, 1.5 years and 5 months) remains transparent and stable.

Since our crystals do not dissolve readily in most organic solvents (except in DMSO), the word "insoluble" was used in the earlier draft. We did not previously perform time dependent solubility tests, but after the reviewer's concern, we have performed dissolution tests for a bunch of crystals in DCM, methanol, acetonitrile, toluene, acetone, and DMSO. We observed that the crystals began to dissolve in DMSO within 8-10 minutes, in acetone within 60 minutes, in DCM within 3 hours. In methanol and acetonitrile, the crystals are sparingly soluble even after 12 hours, and in toluene they are insoluble even after 15 hours.

Time dependent solubility tests on crystal 1 in 6 different solvents.

So, in view of the new observations from solubility tests, we modified the statement and removed the part, "insoluble in most of the organic solvents at ambient conditions" to make the sentence more accurate.

Old text: “The crystals do not degrade even after one year, insoluble in most of the organic solvents at ambient condition and have high thermal stability upto ~ 200 °C (Supplementary Fig. 1), making it a better self-healing material over the previously reported crystals¹.”

New Text: “The crystals of **1** remain stable for more than a year at ambient conditions and show high thermal stability upto ~ 200 °C (Supplementary Fig. 1), making it a better self-healing material over the previously reported crystals¹.”

2. In the Crystal Structure paragraph, it was stated that molecules in **1** are 'connected via strong N–H...O and supportive C–H...O interactions' which 'form a 3D hydrogen-bonded network', and that '...the structure does not have any flat slip planes but has dispersive interaction zones in the crystal packing'. It is important to clarify the two remarks and to distinguish between the two contributions as this will help understand the structural background of this interesting and highly desired phenomenon. This is important especially as the first report on the autonomous self-repair of molecular crystals (ref [1]) claimed that crystals of piezoelectric materials that did not show any evidence of self-healing (the piezoelectric nature of a material 'is not a sufficient condition for self-healing') were comprising 'predominantly strong hydrogen-bonding networks and minimally dispersive interactions'. Also, a clear envisioning/description of the 'criss-cross arrangement' as well as the depiction of the dispersive interaction zones, in addition to HBs (Fig.2), in the crystal packing of **1** would help understanding.

Reply: We thank the reviewer for pointing out this important aspect.

In our earlier report (Ref 1), we mentioned that the application of a mechanical stress on the crystals ruptures the strong hydrogen bonding interactions and generates opposite charges on the two newly created surfaces and the charges remain for significant amount of time due to fatigue or local level plasticity in the vicinity of the fractured tip. The local fatigue or plasticity can be due to the presence of dispersive zones in a crystal structure. Such a combination of interactions is not common in all piezoelectric crystals. We observed (in our earlier work) that crystals with a strong interaction network and minimal dispersive interactions, despite being piezoelectric, do not show attractive surface charges. However, it is too early to conclude that these are indeed the main factors responsible for generation of charges on fracture surfaces. Since our library of self-healing materials is still very limited, we do not want to explicitly associate these (potentially coincidental) structural factors to the successful healing. We are currently gathering more evidence to understanding the mechanism and design principles using both experimental and theoretical studies and will publish them in the near future.

We have made appropriate changes to the text to make it clearer.

Old text:

“The molecules form parallel columns, connected via strong N–H···O (2.03 Å, 163.82°) (amine-to-ester) and supportive C–H···O (2.49 Å, 140.15°) interactions, and form a three dimensional (3D) hydrogen-bonded network (Fig. 2b). Besides, the structure does not have any flat slip planes but has dispersive interaction zones in the crystal packing. Hence the overall density of hydrogen bonds in the structure is moderate with interaction strength comparable in three dimensions (Fig. 2c, d).”

New text:

“The molecules form parallel columns, connected via strong N–H···O (2.03 Å, 163.82°) (amine-to-ester) and supportive C–H···O (2.49 Å, 140.15°) interactions. These hydrogen bonds connect the molecules in the three dimensions (3D) (Fig. 2b). Besides, the structure has dispersive interaction zones in the crystal packing. The overall density of hydrogen bonds in the structure is moderate with interaction strength comparable in three dimensions (Fig. 2c, d). In our earlier study, we reported that the mechanical fracture in some non-centrosymmetric crystals can generate surface charges.¹ Exact structural origins as to why this occurs only in some of the non-centrosymmetric crystals and not in all, is yet to be fully understood.”

3. The exceptional healing ability of cracks (1-3 μm in length) with appreciable healing propensity was claimed, and the authors refer to 'statistical data' of successfully healed crystals. It would be advisable to have the claim supported (in the main text) by numerical data (i.e. the percentage of the healed crystals; the success rate was actually 10-15%). Also, Fig.4 in Supplementary does not present 'statistical data on fracture...', but video grabs of healed (2)/fractured (2)/imperfectly healed crystals (2) out of 20 tested crystals; the figure caption needs to be improved.

Reply: What we meant to infer here was that when we randomly picked, there was a chance that 10-15 % crystals might heal perfectly under gentle mechanical push. As suggested by the referee, we have added the numerical text and data in the MS and SI respectively. New text in manuscript is as below.

New text: “Fracture and healing behaviour under different mechanical stress conditions were studied using about twenty single crystal samples. The analysis revealed that about 2-3 crystals (out of 20) healed perfectly, even when the fracture created cracks larger than ~ 1-3 μm (Supplementary Fig. 4).”

It is worth noting that the success rate can be effectively increased by choosing defect-free crystals without debris as well as application of stress in a more controlled fashion (gentle push or performing three-point bending tests by fixing a crystal on a flexible sheet). We changed the figure caption of Fig. 4 following reviewer's advice as follows,

New caption: "Selected videograbs showing fracture and healing behavior of crystals. Among 20 randomly selected crystals, about 2-3 crystals healed perfectly upon application of a gentle mechanical loading (a, b) while the others either remain shattered (disintegrated pieces) (c, d), or re-joined, producing imperfectly healed state (e, f)."

4. It is not clear how the mechanical fracture tests on crystals of 1 showed that 'they develop long-lasting opposite charges on broken surfaces,...'. This needs clarification. No data on the surface potential of 1 was available nor it was reported that the experiments were performed either.

Thanks for this pertinent question. Indeed, in ref [1], KPFM experiments were performed to measure the surface potential on freshly fractured crystals that showed mechanical actuation. In the present study, the observations (mechanical actuation, piezoelectric space group, ultrafast autonomous self-healing, etc) are very similar, hence we assumed that the same kind of charges are responsible for the mechanical actuation and healing. Since our main focus in this article was to demonstrate the "utility" of the self-healing single crystals, we did not invest much efforts into the self-healing mechanism.

However, following the reviewers' advice, we now performed the Kelvin probe force microscopy (KPFM) measurements on a freshly fractured single crystal of 1. **The test result indeed confirmed that the charges develop on the broken surfaces of crystal, supporting our claims in the current work.** The measurements confirmed that the sample shows an average surface potential of around + 0.68 V, the magnitude of which is in the same order of the values that were found on crystals in Ref [1].

Now, we added a new figure of KPFM experiment in the supplementary information (the new figure and figure caption as below):

Supplementary Fig. 5: KPFM studies on a freshly created surface of (001) of a crystal of 1. (a) Schematic of the preparation of sample for KPFM measurements. Measurement is performed on a freshly created surface of (001) by mechanically fracturing the crystal of **1**, which was mounted vertically using silver paste on a FTO coated glass. (b) Optical image of the AFM cantilever (focused) and freshly created surface of (001) of a crystal of **1** (in out of focus), which was taken prior to performing the experiment. (c) Topography image and (d) surface potential (average surface potential is around + 0.68 V) image of a scan area of $4 \mu\text{m} \times 4 \mu\text{m}$.

5. To present the autonomous actuation, linear acceleration, force and work capacity were calculated from the volume and mass of the crystal portion, and the distance (and time!) the crystal portion travelled. All the data but time are presented in Table 8, Supplementary, which needs to be included too.

Reply: We thank the reviewer for the observation. Now, we included data for time in Table 8.

6. Lns 123, 212; 'enormous' amount of attractive force; it is apparent that the restoring force between the two parts of the fractured crystals was strong indeed, but a term that describes the attraction more realistically would be required.

Reply: Since, the time taken for the autonomous actuation is in milliseconds range, it can be said that large amount of surface charges is responsible for the attraction. Hence, we used the term “enormous” to describe the “strong force of attraction” we observed.

As suggested by the referee, we replaced the word “enormous” with “significant” in the MS.

7. Also, some mostly technical issues need to be resolved as well:

- The 'Results' and 'Discussion' sections; the authors are invited to re-examine the entitling of the sections, especially the latter one which more closely corresponds to Conclusions.

Reply: As per the journal's format, we believe that the current text with modifications is in order.

- Capture to Fig.2e; for consistency reasons, the same face indexing should be used in the figure as in the capture to it; e.g. (100)/(0101) vs. (100). Please do correct it.

Reply: Thanks for the comment. We corrected it in the revised manuscript.

- The ">-shaped" term; although it indeed follows the orientation of the fractured crystal in Figure 3, 'V-shaped' would be better suited and generally acceptable.

Reply: We changed ">-shaped" to 'V-shaped' in revised MS.

- Ln119, '... stereo-microscope equipped with high-speed camera(Fig. 1d-g).' Actually, Fig. 1c-d presents schematics. Please do correct it.

Reply: Thanks for the observation. We corrected it in the revised MS.

- Careful proofreading of the manuscript is required: e.g., Ln 144; '... we measured the time required for the cracks to disappear from the time of withdrawal of the load.', etc.

Reply: Thanks for the suggestions. We proofread the MS and SI files carefully and did necessary changes including the line mentioned above.

- Supplementary, Figs. 5 and 6; is there any need for separating the experiments into two figures and starting Fig. 6 with the label (e) instead of (a)?

Reply: Thanks for pointing out the error. We labelled Fig. 6 (which is now S7), starting with (a) instead of (e) in SI.

Reviewer#2:

Comment:

1. This is an interesting work that reports self-healing in molecular crystals. As an evidence of self healing, authors used SHG but claimed attractive forces at the fractured surface were responsible for self healing. This claim was not reconciled by experimental evidence.

Reply: Thanks to the reviewer for finding our work interesting.

We have added the evidence for the observed surface charges, please see **Comment 4, Reviewer #1**.

2. Authors also need to clarify what they meant by domain structure in diffraction of imperfectly healed crystals.

Reply: In the imperfectly healed crystal, the two crystal pieces are misaligned at the crack-junction. Since each misaligned crystal piece acts as a separate domain, they produce split diffraction spots. Whereas, for healed crystal, due to perfect crystallographic alignment at the healed junction, they generate sharp diffraction spots without any splitting or shoulders. This confirms that the entire crystal acts as a single domain. This is what we tried to explain in the paper. We have revised the text to improve the clarity (the new text is copied below):

New text: “On the other hand, in the imperfectly healed crystal the two crystal pieces are misaligned at the crack-junction. Since each misaligned crystal piece acts as a separate domain, the crystal produces split diffraction spots (Supplementary Fig. 11d).”

Reviewer #3:

Comment:

1. What are the noteworthy results?

The authors report on organic single crystals, which show the ability to heal ruptures over several μm in a fast manner. As claimed by the authors even in unprecedented speed. The structure of the crystals was established by DFT and XRD. The successful healing of crystal "1" was shown by XRD of a single self-healed crystal. The authors also show that optical properties (1 is SHG active material) are fully regained upon self-healing. Thermal stability of the organic crystals is shown by TGA and DSC measurements. Mechanical properties are measured by means of nanoindentation.

Reply: Thanks for appreciating the novelty and quality of our whole work. The take home message is that autonomous self-healing piezoelectric single crystals, owing to their non-centrosymmetry, can find many applications like as robust SHG active materials and ultrafast mechanical actuators as demonstrated in this work.

2. Will the work be of significance to the field and related fields? How does it compare to the established literature? If the work is not original, please provide relevant references. As also pointed out by the authors and shown in an extensive literature research, there is ample use for those properties in applications (autonomous actuators, optical applications). Here, the autonomous self-healing without any additional stimuli and the speed of shape recovery shown by means of a high-speed camera is emphasized and show soignificant progress in this field.

Reply: Yes, this work holds a good promise in this field. And we believe this study may inspire many materials scientists and crystallographers to design future high-end technologies. Based on our knowledge, the crystal structure of **1** is reported for the first time and all the studies reported here are original with several unprecedented observations.

3. Does the work support the conclusions and claims, or is additional evidence needed? The study works as a proof of concept for claimed properties of crystal 1 and hence is in my opinion successful. The authors also show that the behavior of crystal 1 can be repeated in other crystals of the same batch.

Reply: Thank you for appreciating our work. The figures and data are self-explanatory.

4. Are there any flaws in the data analysis, interpretation and conclusions? Do these prohibit publication or require revision? At some points in this study, the authors are not clear on distinguishing a proof of concept from a statistical approach

1) p.5 line 131 to 134 the authors claim to provide statistical data in the Supplement Fig. 4. However, aforementioned Fig. 4 shows a series of images of successful and unsuccessful self-healings and the only mentioning of the success ratio: 2-3 of 20 crystals healed (Supplement line 49-53). This does not suffice as statistical treatment in my opinion.

Reply: Thanks for the comment. Although we performed healing tests on many crystals, in this portion of the text, we attempted to estimate the success rate of healing by randomly picking about 20 crystals for the experiments. We observed that about 2-3 crystals out of 20 can heal precisely while others re-join but leaving a visible crack behind. We provided the images in Fig. 4 in Supplementary information from the representative videograbs to show both the scenarios. In this context, we used the term “statistical” in the main text. We understand that this might be confusing, hence we improved the caption of Figure 4 in SI and also the text in the main draft.

Indeed, as mentioned earlier (see **Comment 3, Reviewer #1**), there are several factors for successful and unsuccessful healing. Gentle push most of the time leads to an efficient healing but as we performed all the tests manually, the outcome depended on many factors like the control on the hand movement, experience of the performer, crystal quality and tools (forceps and needle). We are trying to systematically understand these factors which will be reported in future.

2) In order to describe the phenomenon of the shown behavior of crystal 1 the authors make a very important distinction of self-healing behavior upon “excessive mechanical stress fracture in a brittle manner” (line 119-120) and “gentle force” (line 121). This is unfortunately not picked up in a hypothesis of the mechanism governing the self-healing. Both points were not picked up in the Discussion (from line 341), which reads more as an outlook and does not critically discuss a hypothesis of the self-healing mechanism or the realistic use of this behavior for applications.

Reply: As discussed above, several factors are responsible for a successful healing and there is still a lot to understand. We observed that gentle push normally makes it easier for crystals to align perfectly as compared to an excessive force. The latter often leads to the large separation of the pieces and in such cases the chance of misalignment is much higher. Since

the focus of this work is not on the mechanism of self-healing, we did not go into these details in the text. However, following the suggestion of the referee, we have added an additional explanation in the “Discussion” sections as follows:

New text: “Here, we report a dibenzoate single crystal that is stable in moderate-to-harsh environmental conditions, and which upon gentle stress, is capable of autonomous self-healing in a few milliseconds without needing any external intervention. The crystals of **1** fracture in a brittle manner under excessive mechanical stress, generally leading to imperfect healing, provided the fragments lie within the operational distance of their attractive surface charges.”

5. Is the methodology sound? Does the work meet the expected standards in your field? The authors explicitly state that the focus of this study is not on the mechanical properties of the crystals. And I applaud them for including nanomechanical measurements, which are not trivial with these materials. However, when presenting different mechanisms for material failure and plastic deformation (one brittle without self-healing option and one with fast self-healing option => line 119-121) it should be discussed how this influences mechanical measurements by nanoindentation. An instantaneous self-healing mechanism could make elastic and plastic deformations undistinguishable in the method of Oliver and Pharr. Hence, elasticity and hardness might be overestimated, which might jeopardize the author’s claim that “this material is among the hardest and stiffest autonomous self-healing materials” (line 192).

Reply: We thank the reviewer for raising this interesting point. The indentation experiments normally cause a permanent dent on the crystals beyond the elastic limit of the material. In our case also, we could see a clear indent impression. This means that the deformation by indentation and generation of macroscopic cracks on crystals upon three-point-bending deformation are not the same. Indentation force is generally very high, acting locally as it is applied using a sharp tip (tip radius 150 nm). Indentation etches out or compresses the material under the tip into bulk, so we did not observe any significant self-healing of indent impressions after the indentation measurements. Hence, we believe that the values we obtained using nanoindentation will not have significant effect from the self-healing behaviour. So, our values from nanoindentation provide a good estimation of the real values.

We used the term “this material is among the hardest and stiffest autonomous self-healing materials” only to show how the material is harder and stiffer as compared to other self-healing materials reported in literature. The nanoindentation is done on pristine crystals only. In any case, the E and H values of our crystals are an order of magnitude higher than the most self-

healing materials in literature, involving gels and soft polymers. Hence, our statement is valid in this context.

6. Is there enough detail provided in the methods for the work to be reproduced?
The study is well documented.

Reply: Thank you very much. We have provided detailed description of all methodologies, hence one should be able to easily reproduce our work.

Reviewers' Comments:

Reviewer #2:

Remarks to the Author:

This revised manuscript is acceptable for publication

Reviewer #4:

Remarks to the Author:

We thank the authors for their reply to the reviewer comments. There are, however, several unresolved issues in the revised version, especially on the analysis of data. Based on the previous comments:

4.1 Statistical analysis & reproducibility

The authors analyze the "success rate of healing by randomly picking about 20 crystals for the experiments", and report that only 10% (2-3 out of 20) can be healed while the remaining 90% are damaged with visible cracks. This yield is strikingly low, and it is not clear why only 10% can heal. Are the crystals the same size? Are the forces applied of similar magnitude? Does this lack of reproducibility come from the quality of the crystals or from experimental issues? These are important factors that should be discussed as this has strong implications for the claims of the paper on durability and maintenance on optics applications. For example, with a 10% healing probability, only 1 out of 10 crystals can be repaired after damage, only 1 out of 100 can be repaired after 2 cycles, etc. A systematic analysis under controlled experimental conditions is necessary to support the results.

4.2 Mechanical stress: "excessive" vs "gentle"

The authors report very different healing behavior for crystals under "excessive" versus "gentle" force. This assessment is qualitative, not quantitative, as one cannot differentiate between the applied forces. This is important for two reasons:

a) the healing behavior is drastically different, from perfectly healed with gentle force to imperfectly healed and visible cracks in the crystal with excessive force. This differentiation is important both in terms of properties but also for their realistic application and implementation in nonlinear optic devices, as depending on the forces the device might be repairable or not. The force (or stress) threshold between repairable damage and irreversible damage should be quantified and explicitly reported.

b) without quantitative results, this healing behavior is not reproducible. The authors state that are "several factors for a successful healing", including the movement of the hand, the experience of the performer, quality of tools, etc. These parameters all depend on whoever is performing the experiment, and are therefore subjected to experimental error, human error and bias, and are dependent on the specific equipment and person applying the force. Without a quantitative report on the forces causing the healing or non-healing, the results are not reproducible outside of the authors' lab.

5. best performing claims

The authors claim that "this material is among the hardest and stiffest autonomous self-healing materials" comparing this to other self-healing materials reported in the literature, and refer to Supp. Table 1 to support this claim. Surprisingly, Supp. Table 1 has a very generic comparison with categories such as "polymer", "composite", "hydrogel", etc., with a single reference to the authors' own prior work (Bhunia et al., Science 2021). Given the vast literature and research efforts in the field of self-healing materials, such a reduced comparison is not appropriate to support these claims, and such statements can be misleading. I would encourage the authors to refrain to use such superlative comparisons without the proper in-depth analysis, and would advise to remove such claims from the

article to avoid misleading the readers. After all, the results are already interesting on their own, it doesn't need to be the "best in X" to be something new.

REVIEWER COMMENT

Reviewer#2:

Comments: This revised manuscript is acceptable for publication

Reply: We thank reviewer #2 for appreciating the novelty of our work and recommending it for its publication.

Reviewer #4:

Comments: We thank the authors for their reply to the reviewer comments. There are, however, several unresolved issues in the revised version, especially on the analysis of data. Based on the previous comments:

Reply: We thank the reviewer for his/her efforts to help us identify the gaps for improving the quality of the article further. We addressed all the questions point-by-point below. We are satisfied to share that most of our responses are based on the additional data from new experiments, which we did based on the suggestion by referees. We hope that now the draft is acceptable for publication, in view of the substantial revision, incorporating all the supportive data.

4.1 Statistical analysis & reproducibility

The authors analyze the “success rate of healing by randomly picking about 20 crystals for the experiments”, and report that only 10% (2-3 out of 20) can be healed while the remaining 90% are damaged with visible cracks. This yield is strikingly low, and it is not clear why only 10% can heal. Are the crystals the same size? Are the forces applied of similar magnitude? Does this lack of reproducibility come from the quality of the crystals or from experimental issues? These are important factors that should be discussed as this has strong implications for the claims of the paper on durability and maintenance on optics applications. For example, with a 10% healing probability, only 1 out of 10 crystals can be repaired after damage, only 1 out of 100 can be repaired after 2 cycles, etc. A systematic analysis under controlled experimental conditions is necessary to support the results.

Reply: We humbly point out that the use of the word “reproducibility” by the reviewer here is inappropriate as all our claims are fully reproducible by anyone with basic knowledge of the art. The question is about the “low propensity” for the “perfect-healing” in the crystals. Notably, all the disintegrated crystals can “recombine and heal” but a visible crack remains (imperfect-healing) in many instances. These issues are related to the manual

mechanical loading experiments employed by us which offer low control and provide no quantitative assessment. As suggested by the referee, we conducted new experiments and provided the quantitative data in this version of the draft (Revision-2). We now know the range of forces needed to repeatedly observe the self-healing in crystals (see response below). In our earlier draft, we listed the possible factors that could affect the healing propensity in detail, in order to help the other researchers to reproducibly achieve the results without any difficulty to the level that we claimed.

In the manual three-point bending tests, the surface charge-induced attraction between the fractured shards occurs very quickly, followed by an abrupt closing of the gap between disintegrated shards (within a few milliseconds). The approaching fragments fail to properly realign, hence often fail to achieve perfect healing. It should be noted here that in crystalline solids, healing of even micro- or nano-cracks is extremely rare, and proved to be difficult. Hence, the self-healing of fragmented pieces with “*utmost perfection*” is hardly reported, which reveals the challenge. The most distinct advantage of our crystals is that they are capable of healing with perfection, leaving no macroscopic, observable scars at the fractured position, unlike in most other materials, where not only the cracks remain visible in all instances but also show a compromise on the crystalline integrity at the damaged region, even after hours to days of healing period (vs milliseconds in our case). We mentioned earlier that expertise and sophistication in handling improved our ability to repeatedly demonstrate the “perfect healing” of macroscopic cracks, using the three point bending tests (mentioned in the previous response). It is to be noted that the imperfectly healed crystals also operate as monoliths as they are attracted to one another strongly despite the apparent cracks (in earlier work published by other experts, materials with visible cracks have been considered as “healed”; see here: *Chem. Sci.* **11**, 2606-2613 (2020), *Nat. Chem.* **8**, 618-624 (2016); cited in the manuscript). In this work, we described such crystals’ healing as “imperfectly healed” (cracks remain after removal of stress).

To demonstrate the healing potential of our crystals, we further repeated the experiments by applying uniaxial stress on crystals fixed on a flat support (see Methods, Fig. 2a, b and Supplementary Movies 6, 7). We used a force sensor to quantitatively distinguish the loads required for perfect and imperfect self-healing events (often, the latter is associated with complete fracture of the crystals due to excessive damage). The force sensor with a spherical tip having a diameter of ~190 micrometers was used for compressing the crystals with thicknesses ranging between ~70 to 100 micrometers. As the crystals were fixed on the substrate in this method, we achieved a better control in the experiments and could repeatedly apply load in cycles on each crystal to observe multiple healing cycles (see the Supplementary Movies 6, 7).

Fig. 2: **a** Schematic representation of mechanical loading performed on the crystals of **1** using a force sensor (arrows towards left and right indicate the translation direction of the stage; the green arrow represents a gentle force, which is within the limit for observing crystals' self-healing). **b** Optical image of a crystal under gentle push (load 40 mN; upper panels) and large force (load 74 mN; lower panels). **c** Load vs crystal thickness plot from multiple crystals in loading cycles (green and red symbols correspond to healed and unhealed events under gentle and excessive force, respectively).

Our exhaustive statistical analysis of the fracture tests using this method show that the load required to fracture crystals and achieve perfect self-healing is within milli Newton (mN) range (see Fig. 2c, Supplementary Table 1) for crystals of thickness ranging between ~70 to 100 μm . The loads about ~ 70 mN and beyond mostly resulted in imperfect healing or permanent deformation as this was sufficient to crush the crystals, leaving the material with severe fracture. Nevertheless, using the gentle uniaxial compression stress (typically the loads being < 70 mN), we could show excellent repeatable healing cycles, as high as up to 10 cycles (see Supplementary Movie 8). The controlled uniaxial compression experiments demonstrate that the crystals have an ability to withstand external mechanical stresses repeatedly, as long as the sustainable limit is not crossed.

We removed the Supplementary Fig. 4 and the following statements from the manuscript “Fracture and healing behaviour under different...”, instead we added the above statement from “To demonstrate the healing potential of our crystals....” to Fig. 2 in the manuscript under the section of Results and newly subsection of Quantification of mechanical force for perfect self-healing.

Hence, the results from our uniaxial experiments show an improvement over the results from the three-point-bending experiments. The quantification of loads for healing of different crystals having different thickness has allowed us to successfully identify the difference between excess and gentle force.

4.2 Mechanical stress: “excessive” vs “gentle”

The authors report very different healing behavior for crystals under “excessive” versus “gentle” force. This assessment is qualitative, not quantitative, as one cannot differentiate between the applied forces. This is important for two reasons:

a) the healing behavior is drastically different, from perfectly healed with gentle force to imperfectly healed and visible cracks in the crystal with excessive force. This differentiation is important both in terms of properties but also for their realistic application and implementation in nonlinear optic devices, as depending on the forces the device might be repairable or not. The force (or stress) threshold between repairable damage and irreversible damage should be quantified and explicitly reported.

Reply: We thank the reviewer once again for these suggestions. As detailed above, we employed a force sensor (with a minimum count of 1 mN) to measure the load applied in experiments. We built a small setup to conduct mechanical loading studies (see Methods).

Please check the instrumentation details, Supplementary Movies, and Supplementary Table for the same. To briefly state, we were able to quantify and distinguish between a gentle force and an excess force by manually applying a variety of loads on several crystals. We added a subsection for quantification of load for perfect healing under the results section in the manuscript. According to statistical data, there is a clear limit to the amount of load that crystals of **1** sustain before failing to heal themselves completely (Fig. 2 and Supplementary Table 1). Most notably, repeatability is revealed in the majority of cases (see Supplementary Movies 6,7 and Supplementary Table 1), which supports the potential of our crystals for realistic applications.

b) without quantitative results, this healing behavior is not reproducible. The authors state that are “several factors for a successful healing”, including the movement of the hand, the experience of the performer, quality of tools, etc. These parameters all depend on whoever is performing the experiment, and are therefore subjected to experimental error, human error and bias, and are dependent on the specific equipment and person applying the force. Without a quantitative report on the forces causing the healing or non-healing, the results are not reproducible outside of the authors’ lab.

Reply: This concern has already been addressed above. We believe that these experiments can be replicated by anybody with basic skill in the art and anywhere following our description.

5. best performing claims

The authors claim that “this material is among the hardest and stiffest autonomous self-healing materials” comparing this to other self-healing materials reported in the literature, and refer to Supp. Table 1 to support this claim. Surprisingly, Supp. Table 1 has a very generic comparison with categories such as “polymer”, “composite”, “hydrogel”, etc., with a single reference to the authors’ own prior work (Bhunia et al., Science 2021). Given the vast literature and research efforts in the field of self-healing materials, such a reduced comparison is not appropriate to support these claims, and such statements can be misleading. I would encourage the authors to refrain to use such superlative comparisons without the proper in-depth analysis, and would advise to remove such claims from the article to avoid misleading the readers. After all, the results are already interesting on their own, it doesn’t need to be the “best in X” to be something new.

Reply: We thank the referee for this wise suggestion. We removed the sentence as well as the comparison table (Supplementary Table 1 in previous version of the draft) from manuscript and Supplementary information. Thank you for appreciating our results and finding them interesting.

Reviewers' Comments:

Reviewer #4:

Remarks to the Author:

I thank the authors for revising the manuscript. The new quantitative results on self-healing of the crystals within a specified range of forces are very helpful, and it has improved the article significantly. After these revisions, I do recommend the article for publication.